# Bi-Allelic Novel Variants in *CLIC5* Identified in a Cameroonian Multiplex Family with Non-Syndromic Hearing Impairment

**DOI:** 10.3390/genes11111249

**Published:** 2020-10-23

**Authors:** Edmond Wonkam-Tingang, Isabelle Schrauwen, Kevin K. Esoh, Thashi Bharadwaj, Liz M. Nouel-Saied, Anushree Acharya, Abdul Nasir, Samuel M. Adadey, Shaheen Mowla, Suzanne M. Leal, Ambroise Wonkam

**Affiliations:** 1Division of Human Genetics, Faculty of Health Sciences, University of Cape Town, Cape Town 7925, South Africa; wonkamedmond@yahoo.fr (E.W.-T.); esohkevin4@gmail.com (K.K.E.); smadadey@st.ug.edu.gh (S.M.A.); 2Center for Statistical Genetics, Sergievsky Center, Taub Institute for Alzheimer’s Disease and the Aging Brain, and the Department of Neurology, Columbia University Medical Centre, New York, NY 10032, USA; is2632@cumc.columbia.edu (I.S.); tb2890@cumc.columbia.edu (T.B.); lmn2152@cumc.columbia.edu (L.M.N.-S.); aa4471@cumc.columbia.edu (A.A.); sml3@cumc.columbia.edu (S.M.L.); 3Synthetic Protein Engineering Lab (SPEL), Department of Molecular Science and Technology, Ajou University, Suwon 443-749, Korea; anasirqau@gmail.com; 4West African Centre for Cell Biology of Infectious Pathogens (WACCBIP), University of Ghana, Accra LG 54, Ghana; 5Division of Haematology, Department of Pathology, Faculty of Health Sciences, University of Cape Town, Cape Town 7925, South Africa; shaheen.mowla@uct.ac.za

**Keywords:** non-syndromic hearing impairment, *CLIC5*, Africa

## Abstract

DNA samples from five members of a multiplex non-consanguineous Cameroonian family, segregating prelingual and progressive autosomal recessive non-syndromic sensorineural hearing impairment, underwent whole exome sequencing. We identified novel bi-allelic compound heterozygous pathogenic variants in *CLIC5*. The variants identified, i.e., the missense [NM_016929.5:c.224T>C; p.(L75P)] and the splicing (NM_016929.5:c.63+1G>A), were validated using Sanger sequencing in all seven available family members and co-segregated with hearing impairment (HI) in the three hearing impaired family members. The three affected individuals were compound heterozygous for both variants, and all unaffected individuals were heterozygous for one of the two variants. Both variants were absent from the genome aggregation database (gnomAD), the Single Nucleotide Polymorphism Database (dbSNP), and the UK10K and Greater Middle East (GME) databases, as well as from 122 apparently healthy controls from Cameroon. We also did not identify these pathogenic variants in 118 unrelated sporadic cases of non-syndromic hearing impairment (NSHI) from Cameroon. In silico analysis showed that the missense variant *CLIC5*-p.(L75P) substitutes a highly conserved amino acid residue (leucine), and is expected to alter the stability, the structure, and the function of the CLIC5 protein, while the splicing variant *CLIC5*-(c.63+1G>A) is predicted to disrupt a consensus donor splice site and alter the splicing of the pre-mRNA. This study is the second report, worldwide, to describe *CLIC5* involvement in human hearing impairment, and thus confirms *CLIC5* as a novel non-syndromic hearing impairment gene that should be included in targeted diagnostic gene panels.

## 1. Introduction

Hearing impairment (HI) is the most common sensory disability and is prevalent in about 1 per 1000 live births in high-income countries, with a much higher incidence of up to 6 per 1000 live births in sub-Saharan Africa [1]. When occurring in childhood, HI is associated with impaired language acquisition, learning, and speech development, and affects ~34 million children worldwide (World Health Organisation) [2]. Approximately 30 to 50% of HI cases in Africa have a genetic origin [3,4]. Non-syndromic hearing impairment (NSHI) accounts for about 70% of HI cases of genetic origin and is inherited on an autosomal recessive (AR) mode in approximately 80% of cases [5]. 

Variants in *GJB2* and *GJB6* genes, which are the major contributors to NSHI in Europeans, Asians, and Arabs, are infrequent in most populations of African descent, with a prevalence close to zero [6,7,8]. NSHI is highly genetically heterogeneous [3,4]. To date, about 170 loci and 121 genes have been identified as being associated with NSHI (hereditary hearing loss homepage; Appendix B). Targeted sequencing panels that include >100 HI genes have detected a consistently lower rate of pathogenic and likely pathogenic (PLP) variants in sporadic HI cases of African ancestry, e.g., African Americans (26%), and Nigerians and Black South Africans (4%), compared to >70% for Europeans and Asians [9,10]. However, the detection rate was 70% for 10 mutiplex Cameroonian families [11]. Moreover, the prevalence of autosomal recessive non-syndromic hearing impairment (ARNSHI) pathogenic and likely pathogenic (PLP) variants, using data from the genome aggregation database (gnomAD) database [12] were estimated to account for ARNSHI in 5.2 per 100,000 individuals for Africans/African Americans, compared to 96.9 per 100,000 individuals for Ashkenazi Jews based on sequence data [13]. Therefore, there is an urgent need to investigate HI in populations of African ancestry, particularly multiplex families, using next generation sequencing, to improve knowledge a variants and genes which underlie NSHI in African populations. 

In this study, we generated whole exome sequence (WES) data for samples obtained from a multiplex non-consanguineous Cameroonian family, segregating progressive ARNSHI, and identified novel bi-allelic PLP variants in *CLIC5* in the locus DFNB103. This gene was previously reported to be associated with HI in a single Turkish family [14]. This gene encodes a member of the chloride intracellular channel (CLIC) family of chloride ion channels. The encoded protein associates with actin-based cytoskeletal structures and may play a role in multiple processes including hair cell stereocilia formation, myoblast proliferation, and glomerular podocyte and endothelial cell maintenance. Alternatively, spliced transcript variants encoding multiple isoforms have been observed for this gene (provided by RefSeq). The corresponding mutant mouse model (jbg mouse), which has an intragenic deletion in *CLIC5* resulting in a truncated protein, presents progressive hearing impairment and vestibular dysfunction [15].

## 2. Materials and Methods

### 2.1. Ethics Approval

This study was performed with respect to the Declaration of Helsinki. Ethical approval was granted by the University of Cape Town’s Faculty of Health Sciences’ Human Research Ethics Committee (HREC 484/2019), the Institutional Research Ethics Committee for Human Health of the Gynaeco-Obstetric and Paediatric Hospital of Yaoundé, Cameroon (No. 723/CIERSH/DM/2018), and the Institutional Review Board of Columbia University (IRB-AAAS2343). Written and signed informed consent was obtained from all participants who were 21 years of age or older, and from parents in the case of minors, with verbal assent from participants.

### 2.2. Participants’ Recruitment

The participants’ selection process has been previously reported [16]. The hearing-impaired members of the Cameroonian family (Family 24, Figure 1A) were identified through a community engagement program for the deaf. For all hearing-impaired participants, their detailed personal history and medical records were reviewed by a general practitioner, a medical geneticist, and an ear, nose and throat (ENT) specialist. A general systemic and otological examination was performed, including pure tone audiometry. We followed the recommendation number 02/1 of the Bureau International d’Audiophonologie (BIAP), Belgium.

Genomic DNA samples were extracted from peripheral blood, using the chemagic extraction protocol, in the division of Human Genetics, University of Cape Town, South Africa. Additionally, a group of 118 unrelated Cameroonian individuals living with sporadic NSHI of putative genetic origin (Appendix A) were recruited, to investigate the frequencies of pathogenic variants that could be found. All hearing impaired family members were previously investigated for variants in *GJB2* (through direct sequencing of the entire coding region of *GJB2*), and *GJB6*-D13S1830 deletion (using a multiplex polymerase chain reaction), and were negative [6]. 

A total of 122 ethno-linguistically matched Cameroonian controls without personal or familial history of HI were randomly recruited among blood donors at The Central Hospital of Yaoundé, Cameroon.

### 2.3. Whole Exome Sequencing and Data Analysis

DNA samples from five family members were exome sequenced at Omega Bioservices (Norcross, GA, USA); these samples were obtained from two affected individuals (Figure 1A, II.1, and II.3), their parents (I.1, and I.2), and one unaffected sibling (II.4). Library preparation was performed with an Illumina Nextera Rapid Capture Exome Kit^®^ (Illumina, San Diego, CA, USA) following the manufacturer’s instructions, and the resulting libraries were hybridized with a 37 Mb probe pool to enrich exome sequences. Sequencing was performed on an Illumina HiSeq 2500 sequencer using the pair-end 150 bp run format. Sequencing data were processed using the Illumina DRAGEN Germline Pipeline v3.2.8. Briefly, high-quality reads were aligned to the human reference genome GRCh37/hg19 using the DRAGEN software version 05.021.408.3.4.12, and, after sorting and duplicate marking, variants were called, and individual genomic variant call format (gvcf) files were generated. Joint single nucleotide variant (SNV) and Insertion/Deletion (Indel) variant calling was performed using the genome analysis toolkit (GATK) software v4.0.6.0 [17]. The sex of each individual was verified using plinkv1.9 [18]. Familial relationships for all members were verified via Identity-by-Descent sharing (plinkv1.9) and the Kinship-based INference for Gwas (KING) algorithm [18,19].

### 2.4. Annotation and Filtering Strategy

Variants were annotated and filtered using ANNOVAR [20] and custom scripts. Variants were first prioritized based on the inheritance model, considering both AR and autosomal dominant (AD) modes of inheritance. Subsequently, rare variants with a minor allele frequency (MAF) < 0.005 (for AR) and <0.0005 (for AD) in all populations of the genome aggregation database (gnomAD) were retained. Known pathogenic HI variants listed in ClinVar were also retained, regardless of their frequencies. dbNSFP v3.0 was used to annotate, with 17 bioinformatic tools predicting the deleterious effects of the identified variants [21]. Coding variants were evaluated using Sorting Intolerant from Tolerant (SIFT), polymorphism phenotyping v2 (PolyPhen-2) × 2, MutationAssessor, the likelihood ratio test (LRT), Mendelian clinically applicable pathogenicity (M-CAP) score, Rare Exome Variant Ensemble Learner (REVEL), MutPred, protein variation effect analyzer (PROVEAN), MetaSVM, and MetaLR, while MutationTaster, Eigen, Eigen-PC, functional analysis through Hidden Markov models (FATHMM-MKL), combined annotation dependent depletion (CADD) score, and deleterious annotation of genetic variants using neural networks (DANN) score were used to annotate both coding and non-coding variants [21].

Adaptive boosting (ADA) and random forest (RF) scores derived from dbscSNV v1.1 were used to predict the deleterious effect of variants within splicing consensus regions (−3 to +8 at the 5′ splice site and −12 to +2 at the 3′ splice site) [21,22]. We used phyloP, Genomic Evolutionary Rate Profiling (GERP), SiPhy, and phastCons scores to estimate the evolutionary conservation of the nucleotides and amino acid (aa) residues at which the variants occurred [21,23,24]. The hereditary hearing loss homepage (HHL), online Mendelian inheritance in man (OMIM), human phenotype ontology (HPO), and ClinVar databases were used to determine if there were any existing associations between the identified variants and genes and HI. Candidate variants were considered when: (1) they occurred in known HI genes (and genes expressed in the inner ear); (2) they had a predicted effect on protein function or pre-mRNA splicing (nonsense, missense, start-loss, frameshift, splicing, start-loss, etc.); and (3) they co-segregated with the HI phenotype within the family.

### 2.5. Sanger Sequencing

Sanger sequencing was performed for all the available family members (I.1, I.2, II.1, II.2, II.3, II.4, and II.5; Figure 1A), 118 unrelated sporadic NSHI cases from Cameroon (Appendix A), and 122 apparently healthy controls that were previously recruited as blood donors at The Central Hospital of Yaoundé. Primers to target our variants of interest in exon3 (forward 5′-GAAGGAACATACTGGGGCGA-3′; reverse 5′-AGCGCATTTTTGTTAGGCAGA-3′) and at the exon1-intron1 boundary (forward 5′-CTCTGAGCGAAAGAGAGAAAGAG-3′; reverse 5′-ACTTGTTGCTCCCACGACC-3′) of the *CLIC5* gene were validated using NCBI BLAST. The optimal annealing and extension temperatures for the PCR were 60 °C and 70 °C for 30 s and 1 min, respectively. PCR-amplified DNA products were Sanger sequenced using a BigDye^TM^ Terminator v3.1 Cycle Sequencing Kit and an ABI 3130XL Genetic Analyzer^®^ (Applied Biosystems, Foster City, CA, USA) in the Division of Human Genetics, University of Cape Town, South Africa. Sequencing chromatograms were manually checked using FinchTV v1.4.0, and aligned in UGENE v34.0 to the *CLIC5* reference sequence (ENSG00000112782; retrieved from Ensembl browser).

### 2.6. Evolutionary Conservation of Amino Acids and Secondary Structure Analysis

We performed a multiple sequence alignment (MSA) of human CLIC5 with non-human similar proteins to provide more evidence on the evolutionary conservation of the amino acid residue at which our candidate missense variant occurred. A PSI-BLAST search against the non-redundant protein database of CLIC5 was performed. Non-redundant, non-synthetic CLIC5 proteins from all the different species in the 500 BLAST hits were manually retrieved as FASTA files. The MSA was performed using CLUSTAL Omega v1.2.4 [25] and the MSA file was visualized using Jalview v2.10.5 [26]. Furthermore, PSIPRED v4.0 [27] and Swiss-Model [28] were used to assess the secondary structural features of both protein forms. Additionally, the InterPro [29] database was queried via the InterProScan web service [30] to identify domains and potential domain changes for both protein forms separately.

### 2.7. Protein Modelling

Three-dimensional modelling was performed on the longest isoform of the CLIC5 gene as follows: a homology model of the longest isoform (410 amino acids) of wild-type and mutant CLIC5 [NM_001114086.1: c.701T>C:p.(L234P)] was constructed using the program MODELLER based on the available crystal structure of human chloride intracellular channel protein 5 (PDB ID: 6Y2H) as a template [31]. PYMOL viewer was used for structural visualization and image processing. 

## 3. Results

### 3.1. Participants Phenotypes

A total of seven individuals from “Family 24” were recruited, including three affected individuals (II.1: 36 years old, II.2: 32 years old, and II.3: 25 years old), their parents (I.1: 61 years old, and I.2: 55 years old), and two unaffected siblings (II.4: 18 years old, and II.5: 16 years old) (Figure 1A). The most likely mode of inheritance for the NSHI is AR. From the medical history, no environmental factors were identified as a possible cause of HI, and no HI participant had a history of ophthalmological (blurred or distorted vision, photophobia, eye pain, etc.) or neurological (vertigo, dizziness, etc.) symptoms. Additionally, no vestibular, neurologic, or any other systemic abnormalities were detected by physical examination. A history of prelingual and progressive HI was described for all three affected pedigree members; however, before this study, no formal audiological assessment was performed for any of the family members. Audiological assessment of the three affected individuals revealed bilateral profound sensorineural HI (Figure 1B).

### 3.2. WES Identification of Candidate Gene and Variants

The average target region coverage was about 225×, with 96.30% of the target region being covered to a depth of 10 X or more. After applying our various filtering criteria described in the methods section, two candidate variants were found to occur in a known HI gene (*CLIC5;* MIM:607293) and to co-segregate with the HI phenotype. These two variants which occurred in a compound heterozygous state are the missense variant NM_016929.5:c.224T>C, and the splice-site variant NM_016929.5:c.63+1G>A. The NM_016929.5:c.224T>C variant leads to the substitution of a leucine by a proline amino acid residue at position 75 [NM_016929.5:p.(L75P)] and was predicted to be damaging by 16 of the 17 bioinformatics tools used (Appendix A). The NM_016929.5:c.63+1G>A variant, which occurs in a canonical donor splice site, was predicted damaging by most of the tools that can be used to evaluate non-coding variants, including MutationTaster, FATHMM-MKL, Eigen-PC, CADD, and DANN (Appendix A). Both variants were predicted as occurring in conserved positions of the genome and were both absent from the gnomAD, UK10K, Greater Middle East (GME) variome project databases, as well as the Single Nucleotide Polymorphism Database (dbSNP) (Appendix A). Based on a human splice finder server (HSF v3.1) and NNSPLICE 0.9, the variant NM_016929.5:c.63+1G>A is predicted to break the consensus 5′ donor site “AAGGTAGGT” (which is altered due to the variation “AAGATAGGT“) and probably alter the splicing of the pre-mRNA. The NM_016929.5:c.63+1G>A variant might therefore alter normal protein synthesis and function through various mechanisms. Based on the American College of Medical Genetics’ (ACMG) guidelines for the interpretation of sequence variants, both variants were classified as pathogenic (NM_016929.5:c.63+1G>A: PSV1, PP1-S, PM2, and PP3 and NM_016929.5:c.224T>C: PM2, PP3, PM3, PP1, and PP1-S) [32,33]. In addition to *CLIC5,* only the *CEP250* gene shows compound heterozygous synonymous variants that co-segregate with hearing impairment (Appendix A), which was unlikely to be the cause of the disease.

### 3.3. Sanger Sequencing Confirmation of Variants

Sanger sequencing confirms these candidate variants and their co-segregation with the HI phenotype (Figure 1A,C). The three affected individuals (II.1, II.2, and II.3) were compound heterozygous for both variants, the father (I.1) and an unaffected daughter (II.4) were heterozygous for the missense variant, and the mother (I.2) and the other unaffected daughter (II.5) were both heterozygous for the splice-site variant (Figure 1A). Neither of these variants was detected in the 122 controls or 118 sporadic NSHI cases (Appendix A) from Cameroon.

### 3.4. Analysis of the CLIC5—NM_016929.5(CLIC5):p.(L75P) Variant on the Protein

#### 3.4.1. Evolutionary Conservation of Amino Acids

The NCBI PSI-BLAST search of CLIC5 (NP_058625.2) against the non-redundant protein database found the variant position p.(L75P) to be highly conserved across all non-human species retrieved in the top 500 BLAST hits (Figure 2). As expected, there was substantial conservation across an extensive aa block (on which the variant resides) which forms the thioredoxin/Genetic Diversity Statistics (GST)–N-terminal binding domain. This was consistent with the GERP and PhyloP scores for conservation, indicating a strong evolutionary and functional constraint on the region.

#### 3.4.2. Protein Modelling: Secondary Structure Analysis and Domain Search

A significant attenuation of the protein’s secondary structural features was predicted for the NM_016929.5(*CLIC5*):p.(L75P) variant using the PSIPRED v4.0 server, whereby; there was an abolishment of the β4 strand (Figure 3 and Appendix A red box) and multiple changes affecting the lengths of β strands and several helices were inflicted (Appendix A black boxes). Using Swiss-Model, a similar distortion in the secondary structure of the mutant protein was observed; shortening of the β4 strand, although no β-strand loss was apparent. A domain search with InterProScan (InterPro v80.0) predicted the loss of the N-terminal GST domain due to the variant (Appendix A). This domain loss was also predicted to lead to the abrogation of CLIC5′s protein binding function (GO:0005515). Model parameters were refined and showed improvement in model qualities (Appendix A).

Finally, we performed 3D modelling of the wild-type and mutant long isoform of CLIC5 (Figure 3). The NM_016929.5:c.224T>C missense variant is located in a β-sheet in the extracellular domain of the long isoform of *CLIC5* [NM_001114086.1:c.701T>C:p.(L234P)] (Figure 3c). We found that there was a local perturbation in the hydrophobic interaction of nearby residues at position 234 of the CLIC5 protein (Figure 3d,f). Pro234 affects the shortness of the nearby β-sheet conformation in the mutant protein, as shown in Figure 3f. There was also a difference observed on the surface charge distribution between wild-type and mutant (Figure 3e,g).

## 4. Discussion

This study is, to our knowledge, the first report highlighting the association of HI with *CLIC5* variants in individuals of African ancestry, and the second to demonstrate this association globally. Thus, the data confirms *CLIC5* as a novel HI gene. Both pathogenic variants reported are novel: (NM_016929.5:c.224T>C) and the splicing variant (NM_016929.5:c.63+1G>A), and were not found in 118 unrelated sporadic cases of NSHI cases, reinforcing the genetic and locus heterogeneity nature of HI, and the importance of investigating diverse populations, particularly the understudied African populations, to help to enhance and refine HI disease-gene curation. The contribution of *CLIC5* to NSHI in humans was first described with the identification of a homozygous nonsense variant [NM_016929.5:c.96T>A; p.(Cys32Ter)] that abrogated the protein function and co-segregated with ARNSHI in a Turkish family [14]. The two affected individuals from the aforementioned Turkish family presented an early onset sensorineural HI, which started mildly and progressed to severe-to-profound HI. This HI phenotype is similar to that described in the present study, as our three affected participants described a history of prelingual HI, and presented profound sensorineural HI at the time of the study [14]. The corresponding mutant mice model (*jbg* mice), which has a deletion in the *CLIC5* mice ortholog gene, resulting in impaired hearing and vestibular dysfunction [15]. *CLIC5* was also studied in 69 unrelated Spanish and 50 predominantly Dutch patients with ARNSHI, and no PLP variants were identified [14]. In the present study, we did not find any clinical evidence of vestibular or renal dysfunctions, unlike what was previously reported in the Turkish family [14], as well as in the corresponding mutant mice model (*jbg* mice) that were also shown to have abnormalities in the foot processes of the kidney podocytes leading to proteinuria [34,35]. Biological exploration of the kidney functions of affected Cameroonian individuals with PLP in *CLIC5* should be performed. In addition to the inner ear and kidney abnormalities, the jbg-mutant mice also exhibited emphysema-like lung pathology, hyperactivity, and gastric haemorrhage [14,36]. Additional studies on more families and populations worldwide are needed to refine the phenotype of *CLIC5*-induced HI in humans.

*CLIC5* (mapped on 6p21.1 locus) encodes a protein that belongs to the chloride intracellular ion channel (CLIC) family [37]. The encoded protein (CLIC5) was shown to be highly expressed in the inner ear, and important for sensorineural hearing [15]. CLIC5 protein associates with actin-based cytoskeletal structures and may play a role in multiple processes, including hair cell stereocilia formation [15]. The main function of CLIC5A in the ear is the stabilization of membrane-actin filament linkages at the base of hair cell stereocilia [15]. Therefore, a variant that abrogates CLIC5A or destabilizes its activity would lead to the destabilization of actin-based complexes, fusion, and the elongation of hair cell stereocilia, and consequently, impaired hearing [14,38]. The missense NM_016929.5(*CLIC5*):p.(L75P) variant reported in this study is predicted to lead to the loss of the N-terminal GST domain. This is in turn expected to abrogate CLIC5′s protein binding function (GO:0005515), and is therefore likely to affect binding to ERM proteins. Interaction of CLIC5 with the actin-based cytoskeleton is dependent upon its protein–protein interaction with ERM proteins [38]. 

There are three isoforms of CLIC5 [39]: The canonical isoform CLIC5B (410aa), CLIC5A (251aa) and CLIC5C (205aa). All three isoforms show evidence of expression in the human inner ear, of which CLIC5A shows the highest expression (251aa) [40]. The splice site variant we identified in this study is predicted to affect two of these three isoforms, [NM_016929.5:c.63+1G>A (251 aa; CLIC5A); NM_001256023.1:c.63+1G>A (205 aa; CLIC5C)], including isoform CLIC5A. This splice site variant is located at the 5′ donor canonical splice site of exon 1 of these two isoform transcripts (position +1) and predicted to lead to a loss of the consensus 5′ donor site. The missense variant reported in this study [NM_016929.5: p.(L75P)] is predicted to affect all three isoforms of CLIC5 as a missense change. 

Although the identified variants in the present study are predicted to be pathogenic (Appendix A), and to also affect the structure and function of the protein (Figure 2, Appendix A), more studies in other populations will likely inform and strengthen the HI disease gene-pair curation, globally, as illustrated with this case report.

## 5. Conclusions

We identified bi-allelic novel compound heterozygous pathogenic variants in *CLIC5* (MIM:607293), the missense variant [NM_016929.5:c.224T>C; p.(L75P)] and the splicing variant (NM_016929.5:c.63+1G>A), that co-segregated with non-syndromic autosomal recessive hearing impairment in three affected members of a non-consanguineous family from Cameroon. This study is the second report, worldwide, to describe the *CLIC5*–HI gene-disease pair in humans, and thus confirms *CLIC5* as a novel NSHI that should be included in targeted diagnostic gene panels. Our study emphasizes the urgent need of using WES to investigate hearing impairment in understudied African populations, in order to improve our understanding of hearing pathobiology.

## Figures and Tables

**Figure 1 genes-11-01249-f001:**
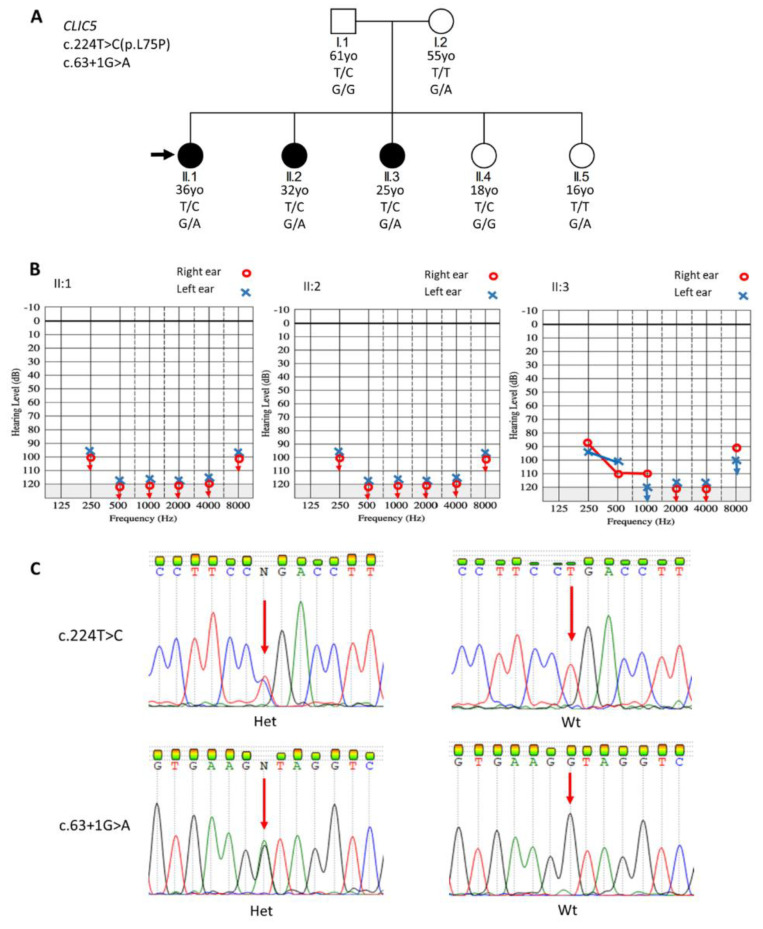
Pedigree of the non-consanguineous family, audiological phenotypes, and electropherogram data of the pathogenic variants in *CLIC5*. (**A**) The pedigree is suggestive of an autosomal recessive mode of inheritance. The missense *CLIC5* variant (NM_016929.5:c.224T>C) and the splicing *CLIC5* variant (NM_016929.5:c.63+1G>A), variants co-segregated with hearing impairment (HI), are compound heterozygous. The black arrow indicates the proband. (**B**) Air conduction of the pure tone audiometry performed for hearing impaired family members. Participants II.1, II.2, and II.3 were presented with a bilateral profound HI. (**C**) Sanger sequencing chromatograms, showing the reference and the alternate alleles of both the missense and the splicing variants. The red arrows indicate the nucleotides affected by the variants. Het, heterozygous for the variant allele; Wt, wild-type (homozygous for the reference allele); yo, years old.

**Figure 2 genes-11-01249-f002:**
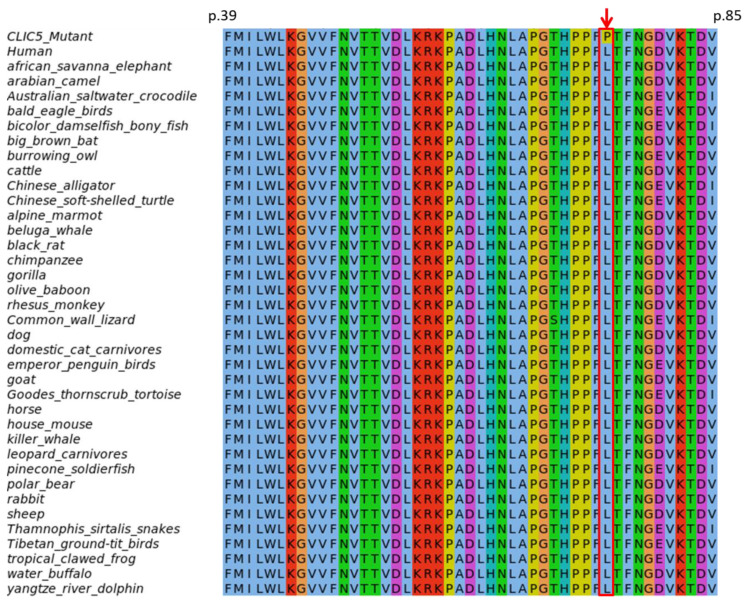
Evolutionary conservation of the *CLIC5*:p.(L75P) variant position (indicated by the red arrow).

**Figure 3 genes-11-01249-f003:**
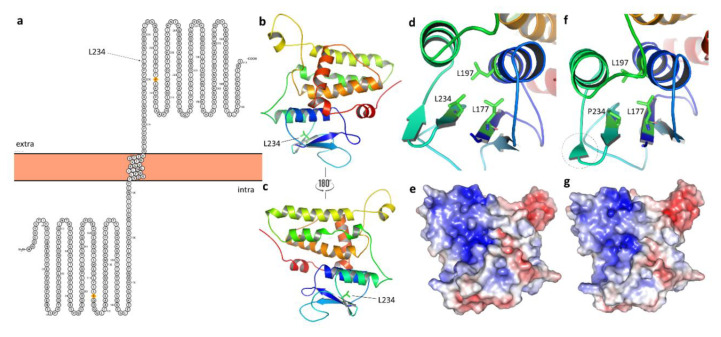
(**a**) The residue Leu234 of NM_001114086.1:c.701T>C:p.(L234P), representing the long isoform of missense variant NM_016929.5:c.224T>C:p.(L75P) is located in the extracellular domain of the CLIC5 protein. (**b**,**c**) The overall structure of CLIC5 and the Leu234 residue (represented by a stick model). (**d**) Close-up view of the interaction pattern at position 234 of wild-type and mutant protein (**f**). Due to the mutation, the shortness of the β-strand observed in the mutant protein was highlighted by a dotted-circle. (**e**) The surface charge distribution of wild and (**g**) mutant CLIC5. Intra: intracellular; extra: extracellular.

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
