# Peer review of "Bi-Allelic Novel Variants in CLIC5 Identified in a Cameroonian Multiplex Family with Non-Syndromic Hearing Impairment"

_genes, 2020, doi:10.3390/genes11111249_

Round 1

Reviewer 1 Report

This is an very well and interesting paper about a family with 3 hearing impaired (profound hearing loss) siblings and normal hearing parents and 2 normal hearing siblings from Cameroon. Authors analyzed WES date from 3 family members, after the previous GJB2 and GJB6 gene testing did not clarified the cause. The onlly gene already associated with hearing loss with autosomal recessive filtering model with biallelic relevant mutations was CLIC5. All three deaf siblings carry both variants , each present in heterozygous state in each parent and i the 2 normal hearing CLIC5 as the hearing impairment causing gene.  Authors also provide theoretical evidence about the impact of the one detected missensemutation on the protein and also in silico analysis of the detected splice site mutation The paper is well writtten and understandable and with a high care. Mutations in CLIC5 seem to be a  very raare cause of AR NSHL, when there are only 2 families wordlwide detected yet since 5 years after the initial paper and when there was no family detected among 69 Spanish and 50 Dutch families with progressive hearing loss.

In the original CLIC5 Tukish family the hearing loss was described also progressive and to profound - so similarly to this reported family - which is also consistent with the interpretation.

There is also a mouse mutatang with mutation in the orthologue gene , having hearinng impairment

So I have only some minor comments.

The abstract shoule be improved , so that it expres the entire messge of the paper. It should be stated in the abstract, that both detected variants were detected in all three deaf or hearing impaired siblings and and only one of them in each hearing sibling and in each parent. And this was done by Sanger sequencing in all 7 family members.

On page 2 , line 91, it should be exaxtly written how were the GJB2 and GJB6 genes tested ?? Was it Sanger sequencing of the entire coding region (exon 2) of GJB2 or only some mutations ? - or MLPA ?

the age of the patients and healthy siblings should be provided in the text in the participants description.

The authors should also show or present which other genes , also genes not yet associated with an human disease (non - OMIM genes) had two sever or biallelic variants ? , or was the CLIC5 the only gene with 2 variants in the affected and each comming from one parent ?? - this would be hardly believable.

It is also statet (page 6, line 208) , that the two novel variants in CLIC5 were submitted to Clinvar under Access Nr  XXX  - the correct number should be provided.

In the discussion the comparison of  hearing loss and the audiological findings in the reported family and the original Turkish family should be discussed and provided

In coclusion on line 293 , the word recessive is lacking after autosomal

Author Response

Reviewer 1:

Reviewer’s comment 1: This is an very well and interesting paper about a family with 3 hearing impaired (profound hearing loss) siblings and normal hearing parents and 2 normal hearing siblings from Cameroon. Authors analyzed WES date from 3 family members, after the previous GJB2 and GJB6 gene testing did not clarified the cause. The onlly gene already associated with hearing loss with autosomal recessive filtering model with biallelic relevant mutations was CLIC5. All three deaf siblings carry both variants , each present in heterozygous state in each parent and i the 2 normal hearing CLIC5 as the hearing impairment causing gene.  Authors also provide theoretical evidence about the impact of the one detected missensemutation on the protein and also in silico analysis of the detected splice site mutation The paper is well writtten and understandable and with a high care. Mutations in CLIC5 seem to be a  very raare cause of AR NSHL, when there are only 2 families wordlwide detected yet since 5 years after the initial paper and when there was no family detected among 69 Spanish and 50 Dutch families with progressive hearing loss.

In the original CLIC5 Tukish family the hearing loss was described also progressive and to profound - so similarly to this reported family - which is also consistent with the interpretation.

There is also a mouse mutatang with mutation in the orthologue gene, having hearinng impairment.

Authors’ response 1: Thank you for the positive feedback.

Reviewer’s comment 2:

So I have only some minor comments.

The abstract shoule be improved, so that it expres the entire messge of the paper. It should be stated in the abstract, that both detected variants were detected in all three deaf or hearing impaired siblings and and only one of them in each hearing sibling and in each parent. And this was done by Sanger sequencing in all 7 family members.

Authors’ response 2: Thank you for your comment. The abstract has been updated, and the section Abstract/P1/L25-30 now reads: “…The variants identified were validated using Sanger sequencing in all seven available family members i.e. the missense [NM_016929.5:c.224T>C; p.(L75P)] and the splicing (NM_016929.5:c.63+1G>A), and co-segregated with HI in the three hearing impaired family members. The three affected individuals were compound heterozygous for both variants, their father and an unaffected sister were heterozygous for the missense variant, and their mother and the other unaffected sister were both heterozygous for the splice-site variant…”

Reviewer’s comment 3: On page 2 , line 91, it should be exaxtly written how were the GJB2 and GJB6 genes tested ?? Was it Sanger sequencing of the entire coding region (exon 2) of GJB2 or only some mutations ? - or MLPA ?

Authors’ response 3: Thank you for your suggestion. The screening methods used for GJB2 and GJB6 genes have been described. The section Materials and Methods/P3/L96-98 now reads: “All hearing impaired family members were previously investigated for variants in GJB2 (through direct sequencing of the entire coding region of GJB2), and the GJB6-D13S1830 deletion (using a multiplex polymerase chain reaction), and were negative”.

Reviewer’s comment 4: the age of the patients and healthy siblings should be provided in the text in the participants description.

Authors’ response 4: Thank you for the comment. The age of all members of “Family 24” have been provided, and the section Results/P6/L184-186 now reads: “A total of seven individuals from “Family 24” were recruited, including three affected individuals (II.1: 36 years old, II.2: 32 years old, and II.3: 25 years old), their parents (I.1: 61 years old, and I.2: 55 years old), and two unaffected siblings (II.4: 18 years old, and II.5: 16 years old)…”

Reviewer’s comment 5: The authors should also show or present which other genes , also genes not yet associated with an human disease (non - OMIM genes) had two sever or biallelic variants ? , or was the CLIC5 the only gene with 2 variants in the affected and each comming from one parent ?? - this would be hardly believable.

Authors’ response 5: Thank you for your comment. We have now provided in the supplementary materials the additional rare filtered variants found which also co-segregate with the hearing impairment phenotype within “Family 24”. Apart from CLIC5, only the CEP250 gene shows compound heterozygous synonymous variants that co-segregate with hearing impairment, which was unlikely the cause of the disease. No rare homozygous variants were found to segregate with the hearing impairment phenotype. The section Supplementary Materials/P9/L313-314 now reads:  “…Table S3: Synonymous likely benign variants identified in the CEP250 gene…”

Reviewer’s comment 6: It is also statet (page 6, line 208), that the two novel variants in CLIC5 were submitted to Clinvar under Access Nr  XXX  - the correct number should be provided.

Authors’ response 6: Thank you for the comment. We submitted both variants to ClinVar prior to submitting the manuscript. Unfortunately, we still did not receive an accession number for them. This sentence (Results/P6/L216: and have been submitted to ClinVar under accession number XXX) has therefore been deleted. We will update the ClinVar submission to include a reference to the manuscript once the manuscript is published. As such, the ClinVar accession number will be linked to the published paper.

Reviewer’s comment 7: In the discussion the comparison of hearing loss and the audiological findings in the reported family and the original Turkish family should be discussed and provided

Authors’ response 7: Thank you for the comment. A comparison of hearing loss and audiological findings in the reported family and the original Turkish family has now been made in the section Discussion/P8/L270-274 which reads: “…The two affected individuals from the aforementioned Turkish family presented with an early onset sensorineural HI which started mildly and progressed to severe-to-profound HI. This HI phenotype is similar to that described in the present study, as our three affected participants described a history of prelingual HI, and presented with profound sensorineural HI at the time of the study…”.

Reviewer’s comment 8: In coclusion on line 293, the word recessive is lacking after autosomal

Authors’ response 8: Thank you for the comment. The word “recessive” has been added, and the section Conclusions/P9/L305-306 now reads: “…that co-segregated with non-syndromic autosomal recessive hearing impairment…”

Reviewer 2 Report

This well written manuscript reports interesting data on two novel pathogenic variants in the CLIC5 gene that co-segregated with non-syndromic autosomal hearing impairment (NSHI) in three affected members of a non-consanguineous family from Cameroon. This study is only second report worldwide supporting the involvement of CLIC5 in human hearing impairment. Moreover, this study is of special interest since it was performed on one of populations of African ancestry which are characterized by very specific genetic contributors to NSHI.

Minor points:

- Lines 184-185: Authors report that “A history of prelingual and progressive HI was described for all three affected pedigree members”. However, it would be interesting to know if there is more detailed information about the pathological phenotypes of affected family members – their hearing loss level at the disease onset and presumed HL progression to profound hearing loss observed at time of examination?

- Line 205:  “…both variants … have been submitted to ClinVar under Accession number XXX.” - Accession number is not presented.

- References 40, 41, 42 are absent in the list of References.

Author Response

Reviewer 2:

Reviewer’s comment 1: This well written manuscript reports interesting data on two novel pathogenic variants in the CLIC5 gene that co-segregated with non-syndromic autosomal hearing impairment (NSHI) in three affected members of a non-consanguineous family from Cameroon. This study is only second report worldwide supporting the involvement of CLIC5 in human hearing impairment. Moreover, this study is of special interest since it was performed on one of populations of African ancestry which are characterized by very specific genetic contributors to NSHI.

Authors’ response 1: We thank you for the positive comment.

Minor points:

Reviewer’s comment 2: - Lines 184-185: Authors report that “A history of prelingual and progressive HI was described for all three affected pedigree members”. However, it would be interesting to know if there is more detailed information about the pathological phenotypes of affected family members – their hearing loss level at the disease onset and presumed HL progression to profound hearing loss observed at time of examination?

Authors’ response 2: Thank you for your comment. Unfortunately, prior to this study, no formal audiological assessment was performed for any of the three affected members of “Family 24”. We only found a history of early onset and progressive hearing impairment, however, we do not have documentary evidence of the level of hearing loss at onset of the disease nor at different ages. A pressicion to clarify this has now been added to the section Results/P6/L192-193: “…however, before this study, no formal audiological assessment was performed for any of them…”

Reviewer’s comment 3: - Line 205:  “…both variants … have been submitted to ClinVar under Accession number XXX.” - Accession number is not presented.

Authors’ response 3: Thank you for the comment. We submitted both variants to ClinVar prior to submitting the manuscript. Unfortunately, we still did not receive an accession number for them. This sentence (Results/P6/L216: and have been submitted to ClinVar under accession number XXX) has therefore been deleted. We will update the ClinVar submission to include a reference to the manuscript once the manuscript is published. As such, the ClinVar accession number will be linked to the published paper.

Reviewer’s comment 4: - References 40, 41, 42 are absent in the list of References.

Authors’ response 4: Thank you dear reviewer for your comment. References 40, 41, and 42 have now been updated to “37, 38, and 39”.

Reviewer 3 Report

The paper by Wonkam-Tingang entitled: Bi-allelic novel variants in CLIC5 identified in a Cameroonian multiplex family with non-syndromic hearing impairment identify a novel bi-allelic compound heterozygous pathogenic variants in CLC% using whole exome-sequencing and validated by Sanger sequencing. This is the second report to describe CLIC5 involvement in human hearing impairment.

Abstract : well written stating the problem and objectives

Introduction:  The introduction is well developed pertinent background bibliography and summarizing the purpose of the study.

Material and methods: detailed description of methods

Results and figures: authors describe their own finding and supportive images.

Discussion and conclusions: Contrast the results with previous findings and acknowledge the limitations.

Bibliography : comprehensive and balanced for the manuscript content.

Author Response

Reviewer 3:

Reviewer’s comment 1:

The paper by Wonkam-Tingang entitled: Bi-allelic novel variants in CLIC5 identified in a Cameroonian multiplex family with non-syndromic hearing impairment identify a novel bi-allelic compound heterozygous pathogenic variants in CLC% using whole exome-sequencing and validated by Sanger sequencing. This is the second report to describe CLIC5 involvement in human hearing impairment.

Abstract : well written stating the problem and objectives

Introduction:  The introduction is well developed pertinent background bibliography and summarizing the purpose of the study.

Material and methods: detailed description of methods

Results and figures: authors describe their own finding and supportive images.

Discussion and conclusions: Contrast the results with previous findings and acknowledge the limitations.

Bibliography: comprehensive and balanced for the manuscript content.

Authors’ response 1: We are grateful dear reviewer for the positive comment

Other changes:

-P1/L5-6: the author’s name “Liz M Noel” has been updated to “Liz M. Nouel-Saied”

-Author contributions/P9/L320: the abbreviated author name L.M.N. has been updated to “L.M.N.S.”